# Mapping Plant Nitrogen Concentration and Aboveground Biomass of Potato Crops from Sentinel-2 Data Using Ensemble Learning Models

**Hang Yin [1], Fei Li [1,*], Haibo Yang [1], Yunfei Di [1], Yuncai Hu [2] and Kang Yu [2]**

[1] Inner Mongolia Key Laboratory of Soil Quality and Nutrient Resource, College of Grassland, Resources and Environment, Inner Mongolia Agricultural University, Hohhot 010011, China; hangyin@emails.imau.edu.cn (H.Y.); haiboyang@imau.edu.cn (H.Y.); diyunfei2015@163.com (Y.D.)

[2] Department Life Science Engineering, School of Life Sciences, Technical University of Munich, 85354 Freising, Germany; yc.hu@tum.de (Y.H.); kyu@uni-koeln.de (K.Y.)

* Correspondence: lifei@imau.edu.cn

**Abstract:** Excessive nitrogen (N) fertilization poses environmental risks at regional and global levels. Satellite remote sensing provides a novel approach for large-scale N monitoring. In this study, we evaluated the performance of different types of spectral bands and indices (SIs) coupled with ensemble learning models (ELMs) at retrieving the plant N concentration (PNC) and plant aboveground biomass (AGB) of potato from Sentinel-2 images. Cloud-free Sentinel-2 imagery was acquired during the tuber-formation to starch-accumulation stages from 2020 to 2021. Fourteen optimal SIs were selected using the successive projections algorithm (SPA) and principal component analysis (PCA). The PNC and AGB estimation models were then built using an ELMs. The results showed that the SIs based on chlorophyll absorption bands were strongly related to potato PNC and AGB. Also, the N-correlated bands were mainly concentrated in the red-edge (705 nm) and short-wave infrared (1610 and 2190 nm) regions. The ELMs successfully predicted PNC and AGB ($R^2_{PNC} = 0.74$; $R^2_{AGB} = 0.82$). Compared with the other five base models (k-nearest neighbor (KNN), partial least squares regression (PLSR), support vector regression (SVR), random forest (RF), and Gaussian process regression (GPR)), the ELMs provided higher PNC and AGB estimation accuracy and effectively reduced overfitting to training data. This study demonstrated that the promising solution of using SPA-PCA coupled with an ensemble learning model improves the estimation accuracy of potato PNC and AGB based on Sentinel-2 imagery data.

**Keywords:** ensemble learning model; feature selection; plant nitrogen concentration; spectral indices; potato; Sentinel-2 imagery





## 1. Introduction

Nitrogen (N) is essential for higher levels of photosynthetic and plant productivity. Large amounts of N fertilizer are applied to fields to maintain high yields [1–3]. However, soil with excessive nitrogen can lead to significant environmental problems such as nitrogen gasification, leaching, and runoff [4,5]. Plant N concentration (PNC) and aboveground biomass (AGB) information play crucial roles in determining the N nutrition index (NNI) and guiding N fertilization management [6,7]. In addition, PNC and AGB reflect the growing condition of crops and can effectively reflect the final yield. Therefore, the accurate and rapid acquisition of PNC and AGB is essential for optimal management and yield forecasting at regional and global scales [8].

Satellite remote sensing is a highly effective approach for predicting PNC and AGB at both field and regional scales. The predominant technique involves utilizing spectral indices (SIs) to retrieve crucial crop variables [9–11]. However, broad-band SIs can lose their sensitivity as a result of proportionally extensive soil backgrounds or large biomass [12,13].

Therefore, several SIs have been suggested for estimating crop parameters using satellite data, such as the Optimized Soil-Adjusted Vegetation Index (OSAVI) and the Transformed Chlorophyll Absorption Reflectance Index (TCARI) [13,14]. Moreover, Sentinel-2 imagery comprises three red-edge bands, enabling the effective prediction of crop PNC and AGB [15,16]. Some studies have demonstrated that SIs including the red-edge region improved the estimation accuracy of plant parameters [17,18]. Gitelson [19] developed the red-edge chlorophyll index ($CI_{rededge}$), which effectively predicted the chlorophyll content of corn and soybeans. However, despite 13 bands of Sentinel-2 imagery, few studies analyze which types of SIs and which combination of bands better predict potato PNC and AGB.

Unlike SIs, machine learning algorithms (MLAs) estimate crop parameters through explicit regression functions and can effectively explore optical data [4,20]. Therefore, many studies have combined SIs with MLAs to enhance the accuracy of crop PNC and AGB estimation [21,22]. However, the potential multicollinearity among SIs still impacts the estimation performance of MLAs [23]. To solve this problem, many dimensionality reduction methods have been proposed [24,25]. One is the feature extraction method, which transforms the original features into spaces, enabling the resulting low-dimensional features to encompass most information, as exemplified by techniques such as principal component analysis (PCA) [25]. However, the data from all principal components (PCs) are less clear than the original information. Another method involves feature selection, where a subset of features is chosen to retain the essential information from the original features, such as the successive projections algorithm (SPA) [26]. Therefore, in this study, we conjecture that combining two dimensionality reduction methods can further reduce the effect of multicollinearity.

Currently, many SIs have been employed to predict crop parameters at large scales. In addition, Sentinel-2 imagery, which involves three red-edge bands, has been widely used to estimate plant parameters. However, only limited research work has explored the potential of multiple types of SIs and different combinations of bands based on Sentinel-2 imagery for potato PNC and AGB estimation. Therefore, the objectives of this study were as follows: (1) to identify sensitive bands for predicting PNC and AGB and determining quantitative relationships between PNC and AGB and different types of SIs; (2) to evaluate different types of SIs for estimating potato PNC and AGB with different models (KNN, PLSR, SVR, RF, GPR, and ELM) and explore the performance of different methods (SPA, PCA, and SPA-PCA) at PNC and AGB estimation; and (3) map the spatial–temporal variability of potato PNC and AGB at different growth stages and validate prediction map accuracy.

## 2. Materials and Methods

### 2.1. Experimental Design

The study was conducted in Zhuozi and Wuchuan counties in the central region of Inner Mongolia, China, from June to September (Figure 1). The potato growing season is mainly concentrated from June to September. The average annual precipitation is approximately 350 mm, 90–95% of which occurs from April to October. Throughout potato growth, average temperature ranges from 20 to 25 °C.

Two potato varieties, Yingniweite and 226, were used under different N fertilizer treatments. Forty-four plots were planted with an area of 120 m$^2$. Experiment 1 had five N levels: three optimized N fertilization application algorithms proposed by Sripada [27,28], Holland [29,30], and van Evert [31]; a N fertilizer optimization algorithm based on N balance combined with spectral indices; and local farmers' practices at 198, 204, 264, 313, and 373 kg N ha$^{-1}$, respectively. Each treatment had four replicates. Experiment 2 had six N levels: (i) control (no N was applied); (ii) integrated management-based N fertilizer application algorithm at 60 kg N ha$^{-1}$; (iii) soil-test-based N fertilizer application algorithm at 80 kg N ha$^{-1}$; (iv) spectral-index-based N fertilizer optimization algorithm at 151 kg N ha$^{-1}$; (v) N-balance-based N fertilizer optimization algorithm at 238 kg N ha$^{-1}$; and (vi) local farmers' practice at 325 kg N ha$^{-1}$. There were four replicates for each treatment. Drip irrigation was applied in both experiments.

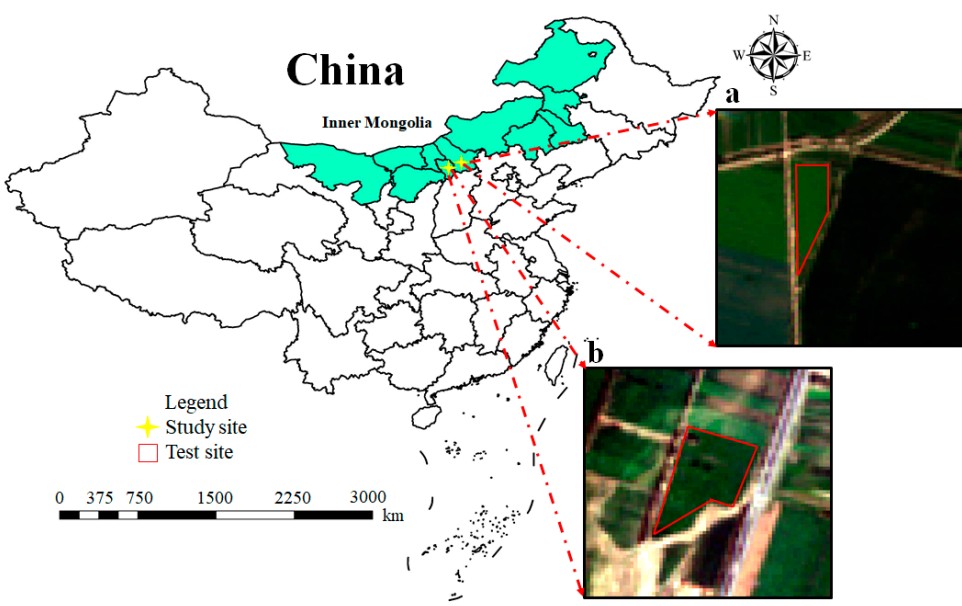

**Figure 1.** Experimental design at two study sites: (**a**) at Zhuozi in 2020 and (**b**) at Wuchan in 2021.

*2.2. Plant Nitrogen Concentration and Aboveground Biomass Measurements*

Plant samplings were taken on 19 July 2020, 10 August 2020, 20 August 2020, 23 July 2021, 12 August 2021, and 29 August 2021, which corresponded to periods of potato tuber formation (T1), tuber bulking (T2), and starch accumulation (T3), respectively. Potato requires large amounts of N nutrients to meet the demand of tuber formation and bulking during the tuber-formation and tuber-bulking stages [32,33]. In addition, precise monitoring of PNC and AGB during the starch-accumulation stages can ensure the rational application of N fertilizer at the late reproductive growth stages. Therefore, the accurate monitoring of PNC and AGB for the three critical growth stages is very important. The aboveground plants of two 1 m consecutive rows of potato of each test plot were sampled. The sample was chopped and mixed, and a 400–600 g sample was taken. The samples were dried at 70 °C and subsequently weighed and chemically analyzed. The Kjeldahl-N method was used to measure the potato PNC. AGB can be computed using the following Equation (1). Samples obtained from each plot were used to represent the changes in potato PNC and AGB for each test plot. PNC and AGB are summarized in Table 1.

$$\text{AGB}\left(\text{t ha}^{-1}\right) = \frac{\text{FW}}{1+\text{M}}/1.8 \times 10000 \tag{1}$$

where FW is the fresh weight of all the aboveground plants (kg) and M is the water content of potato plants (%).

**Table 1.** Summary of PNC and AGB of potato.

| Stages | Number | Min | Max | Mean | SD |
|---|---|---|---|---|---|
| Plant nitrogen concentration (%) | | | | | |
| Tuber formation (T1) | 44 | 3.33 | 4.66 | 4.08 | 0.33 |
| Tuber bulking (T2) | 44 | 2.64 | 4.28 | 3.47 | 0.40 |
| Starch accumulation (T3) | 44 | 1.54 | 3.97 | 2.84 | 0.56 |
| Calibration | 92 | 1.54 | 4.66 | 3.51 | 0.67 |
| Validation | 40 | 1.80 | 4.64 | 3.36 | 0.65 |
| Plant aboveground biomass (t ha$^{-1}$) | | | | | |
| Tuber formation (T1) | 44 | 0.50 | 2.37 | 1.29 | 0.60 |
| Tuber bulking (T2) | 44 | 1.39 | 4.08 | 2.56 | 0.57 |
| Starch accumulation (T3) | 44 | 1.39 | 4.79 | 2.98 | 0.85 |
| Calibration | 92 | 0.50 | 4.79 | 2.23 | 1.00 |
| Validation | 40 | 0.60 | 3.93 | 2.39 | 0.98 |

### 2.3. Sentinel-2 Images Acquisition

Sentinel-2A images of the potato crops acquired on 14 July 2020, 3 August 2020, 16 August 2020, 19 July 2021, 13 August 2021, and 31 August 2021 were utilized. Clouds did not cover the Sentinel-2 images on these dates. The Sentinel-2A level product was acquired from the ESA's Copernicus. The center positions and spatial resolutions of all bands from the Sentinel-2 images are summarized in Table 2. All bands were resampled to 10 m spatial resolution using the resampling tool in the Sentinel Applications Platform (SNAP) software 9.0 and the nearest method. This study utilized ten bands (490, 560, 665, 705, 740, 783, 842, 865, 1610, and 2190 nm) from the Sentinel-2 images. The other three bands (443, 945, and 1375 nm) are used for image atmospheric correction, so they were not used in this study. The coordinates of each test plot were imported from the Sentinel-2 images, and then the average reflectance of each test plot was extracted using the region of interest (ROI) tool in ENVI 5.6 software for subsequent data analysis.

**Table 2.** Summary of Sentinel-2 image data.

| Band | Band Name | Center Wavelength | Bandwidth (nm) | Ground Resolution (m) |
|---|---|---|---|---|
| B1 | Coastal aerosol | 443 | 20.00 | 60 |
| B2 | Blue | 490 | 65.00 | 10 |
| B3 | Green | 560 | 35.00 | 10 |
| B4 | Red | 665 | 30.00 | 10 |
| B5 | RE1 | 705 | 15.00 | 20 |
| B6 | RE2 | 740 | 15.00 | 20 |
| B7 | RE3 | 783 | 20.00 | 20 |
| B8 | NIR1 | 842 | 115.00 | 10 |
| B8a | NIR2 | 865 | 20.00 | 20 |
| B9 | Water vapour | 945 | 20.00 | 60 |
| B10 | SWIR-cirrus | 1375 | 30.00 | 60 |
| B11 | SWIR1 | 1610 | 90.00 | 20 |
| B12 | SWIR2 | 2190 | 180.00 | 20 |

RE: Red-edge band; NIR: near infrared band; SWIR: short-wave infrared.

### 2.4. Spectral Indices Calculation

Spectral indices (SIs) have been widely used to estimate plants' critical parameters. To evaluate the performance of SIs in retrieving potato PNC and AGB, 14 SIs commonly used in the literature to predict crop PNC and AGB were selected in this study (Table 3). SIs are divided into several categories based on type of application, such as chlorophyll content, nitrogen concentration, vegetation, and biomass. The bands of published spectral indices (Table 3) may not apply to predicting potato crop PNC and AGB due to differences in crop species and growth stages [34,35]. Therefore, in this study, we optimized the sensitive bands based on the formulas of published spectral indices. According to the formulas in Table 3, we combined all bands of Sentinel-2 for calculation, thus re-selecting the optimal bands for each published spectral index.

**Table 3.** The spectral indices used in this work.

| Abbreviation | Formulas | Algorithms | Variable | References |
|---|---|---|---|---|
| NDVI | (NIR − Red)/(NIR + Red) | $(R_{\lambda 1} - R_{\lambda 2})/(R_{\lambda 1} + R_{\lambda 2})$ | Biomass/Others | [36] |
| RVI | NIR/Red | $R_{\lambda 1}/R_{\lambda 2}$ | Vegetation | [37] |
| DVI | NIR − Red | $R_{\lambda 1} - R_{\lambda 2}$ | Vegetation | [38] |
| $CI_{red\ edge}$ | (NIR/Green) − 1 | $(R_{\lambda 1}/R_{\lambda 2}) - 1$ | Chlorophyll/LAI | [19] |
| OSAVI | 1.16 × (NIR − Red)/(NIR + Red + 0.16) | $1.16 \times (R_{\lambda 1} - R_{\lambda 2})/(R_{\lambda 1} + R_{\lambda 2} + 0.16)$ | Vegetation | [39] |
| MTCI | (Rededge2 − Rededge1)/(Rededge1 − Red) | $(R_{\lambda 1} - R_{\lambda 2})/(R_{\lambda 2} - R_{\lambda 3})$ | Chlorophyll | [40] |
| MCARI | [(Rededge1 − Red) − 0.2 × (Rededge1 − Green)] × (Rededge1/Red) | $[(R_{\lambda 1} - R_{\lambda 2}) - 0.2 \times (R_{\lambda 1} - R_{\lambda 3})] \times (R_{\lambda 1}/R_{\lambda 2})$ | Chlorophyll | [12] |
| PSRI | (Red − Green)/Rededge2 | $(R_{\lambda 1} - R_{\lambda 2})/R_{\lambda 3}$ | Vegetation | [41] |
| mSR705 | (Rededge2 − Blue)/(Rededge1 − Blue) | $(R_{\lambda 1} - R_{\lambda 2})/(R_{\lambda 3} - R_{\lambda 2})$ | Chlorophyll | [42] |

**Table 3.** *Cont.*

| Abbreviation | Formulas | Algorithms | Variable | References |
|---|---|---|---|---|
| mND705 | (Rededge2 − Blue)/(Rededge2 + Rededge1 − 2 × Blue) | $(R_{\lambda 1} - R_{\lambda 2})/(R_{\lambda 1} + R_{\lambda 2} - 2 \times R_{\lambda 3})$ | Chlorophyll | [42] |
| TCARI | 3 × [(Rededge1 − Red) − 0.2 × (Rededge1 − Green) × (Rededge1/Red)] | $3 \times [(R_{\lambda 1} - R_{\lambda 2}) - 0.2 \times (R_{\lambda 1} - R_{\lambda 3}) \times (R_{\lambda 1}/R_{\lambda 2})]$ | Chlorophyll | [13] |
| NPDI | $(CI_{rededge} - CI_{rededge\ MIN})/(CI_{rededge\ MAX} - CI_{rededge\ MIN}$ | $(CI_{rededge} - CI_{rededge\ MIN})/(CI_{rededge\ MAX} - CI_{rededge\ MIN})$ | Nitrogen | [43] |
| MCARI/OSAVI | MCARI/OSAVI | MCARI/OSAVI | Chlorophyll | [44] |
| TCARI/OSAVI | TCARI/OSAVI | TCARI/OSAVI | Chlorophyll | [13] |

R: the abbreviation of reflectance; λ: the wavebands of spectral indices.

### 2.5. SPA and PCA

The multicollinearity among the input variables significantly affected the accuracy of the crop parameter prediction model. In this study, the SPA and PCA were used to reduce the dimensionality of input variables. The SPA is a mathematical technique used for dimensionality reduction and feature extraction that operates by projecting high-dimensional data onto a lower-dimensional space while preserving specific properties or structures of the original data. PCA is a statistical technique employed for dimensionality reduction and data transformation. It identifies the principal components or axes along which the data varies the most. These principal components are orthogonal, and the transformation results in a new set of uncorrelated variables called main components. PCA is widely used in various fields to simplify data while retaining essential information, making it a powerful tool for exploratory data analysis and feature extraction. The SPA combined with PCA was also employed in this study to reduce the dimensionality of the input variables. For a detailed description of the SPA and PCA according to Fan [45] and Howley [46] see their respective studies.

### 2.6. Machine Learning Model Construction

The three main types of ensemble learning include bagging, boosting, and stacking. Both bagging and boosting, such as random forest (RF) and extreme gradient boosting (XGBoost), are homogeneous learners. Unlike RF and XGBoost, stacking integrates different learners. The stacking ensemble learning that is widely used in regression prediction research generally includes two levels. The essential learners consist of multiple machine learning models that generate meta-feature datasets by learning the original datasets, and the meta-learner produces the final result by learning the meta-feature datasets [47]. This is because 5-fold cross-validation is commonly applied at two levels to ensure a fair and comprehensive comparison of different models and their input variables. In this study, therefore, the essential learners included the k-nearest neighbor (KNN) [48], partial least squares regression (PLSR) [49] to support vector regression (SVR) [50], random forest (RF) [34], and Gaussian process regression (GPR) [51]. At the same time, the new model used multiple linear regression (MLR). The ensemble learning model was performed using the "StackingRegressor" library in Python 10.3. The procedures are shown in Figure 2. In addition, this study compared the prediction effects of the three ensemble strategies (RF, XGBoost, and stacking model) for potato PNC and AGB [52].

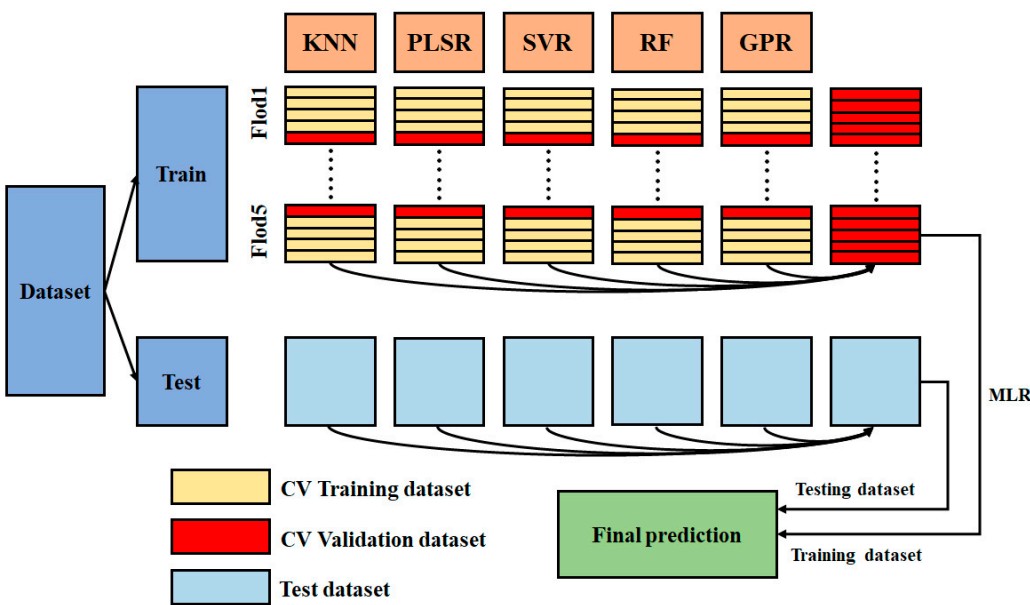

**Figure 2.** The workflow of the ensemble learning model for PNC and AGB estimation.

*2.7. Model Performance Evaluation*

The workflow of the present study is shown in Figure 3. Three accuracy assessment metrics—coefficient of determination ($R^2$), root mean square error (RMSE), and the ratio of performance to deviation (RPD)—were employed to evaluate the performance of all models.

$$R^2 = \sum(y_i - \overline{y})^2 / \sum(y_i - \hat{y}_i)^2 \tag{2}$$

$$RMSE = \sqrt{\frac{1}{n}\sum_{i=1}^{n}(y_i - \hat{y}_i)^2} \tag{3}$$

$$RPD = \frac{SD}{RMSE} \tag{4}$$

where $\hat{y}_i$, $y_i$, and $\overline{y}$ are the measured, predicted, and mean values of PNC and AGB, respectively, n is the number of samples, and SD is the standard deviation of the reference values.

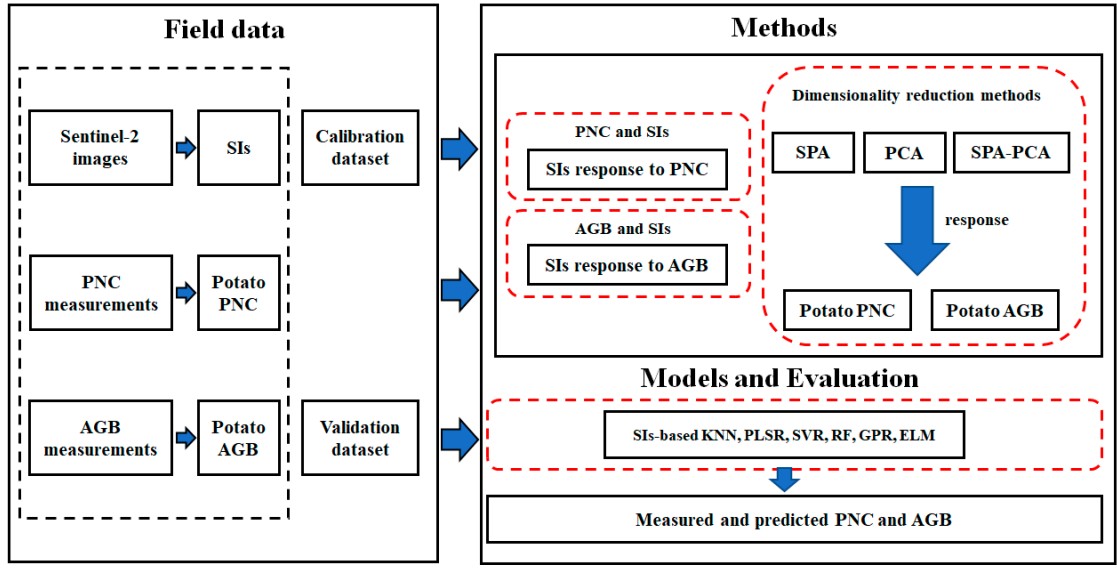

**Figure 3.** Flowchart of potato PNC and AGB estimation modeling methods.

Lin's concordance correlation coefficient (LCCC) quantifies the deviation of the predicted values from a unity line (1:1), representing the extent to which the predictions align with a concordance line of slope 1.0 through the origin. It serves as a measure of the agreement between the predicted and observed outcomes [53]. The LCCC was calculated as per the following equation:

$$LCCC = \frac{2pa_x a_y}{a_x^2 + a_y^2 + (b_x - b_y)^2} \tag{5}$$

where $b_x$ and $b_y$ are the measured and predicted PNC means, respectively; $a_x^2$ and $a_y^2$ are the variances of measured and predicted PNC and AGB; and p is the Pearson correlation coefficient between measured and predicted PNC and AGB.

## 3. Results

### 3.1. Correlation Analysis of SIs and PNC and AGB

According to Table 3, all possible band combinations were used to analyze the Sentinel-2 bands using the training dataset. The best-performing bands per formulation are listed in Table 4. The optimal sensitive bands were different between PNC and AGB. For PNC, sensitive bands were obtained in the RE (705 nm), NIR (865 nm), and SWIR (1610 and 2190 nm) regions. For AGB, sensitive bands were obtained in the visible light regions (490 and 560 nm), and NIR (842 nm), and SWIR regions (1610 and 2190 nm) (Table 4).

**Table 4.** Optimal bands for estimating PNC and AGB of potato.

| Spectral Indices | PNC | | | | AGB | | | |
|---|---|---|---|---|---|---|---|---|
| | $R_{\lambda 1}$ | $R_{\lambda 2}$ | $R_{\lambda 3}$ | $R^2$ | $R_{\lambda 1}$ | $R_{\lambda 2}$ | $R_{\lambda 3}$ | $R^2$ |
| NDVI | 705 | 2190 | | 0.53 | 490 | 842 | | 0.39 |
| RVI | 705 | 2190 | | 0.49 | 490 | 842 | | 0.33 |
| DVI | 705 | 1610 | | 0.65 | 560 | 1610 | | 0.56 |
| CI$_{red\ edge}$ | 705 | 2190 | | 0.49 | 490 | 842 | | 0.33 |
| OSAVI | 490 | 2190 | | 0.56 | 560 | 1610 | | 0.40 |
| MTCI | 705 | 865 | 1610 | 0.62 | 842 | 1610 | 2190 | 0.40 |
| MCARI | 705 | 705 | 1610 | 0.65 | 560 | 560 | 1610 | 0.57 |
| PSRI | 865 | 1610 | 2190 | 0.57 | 842 | 1610 | 2190 | 0.40 |
| mSR705 | 705 | 865 | 1610 | 0.62 | 842 | 1610 | 2190 | 0.40 |
| mND705 | 490 | 560 | 2190 | 0.60 | 490 | 842 | 1610 | 0.47 |
| TCARI | 705 | 705 | 1610 | 0.65 | 560 | 705 | 1610 | 0.59 |
| NPDI | 705 | 1610 | 2190 | 0.62 | 490 | 783 | 842 | 0.40 |
| MCARI/OSAVI | 705 | 865 | 1610 | 0.65 | 560 | 705 | 1610 | 0.59 |
| TCARI/OSAVI | 705 | 865 | 1610 | 0.65 | 560 | 705 | 1610 | 0.59 |

R: the abbreviations of reflectance; λ: the wavebands of spectral indices; $R^2$: the coefficient of determination.

To determine the relationship between PNC and AGB and spectral indices, these two variables were correlated with SIs (Table 4). Most SIs were positively related to PNC and AGB. Compared to other types of SIs, SIs based on chlorophyll content were strongly correlated to the PNC and AGB of potato. The coefficient of determination ($R^2$) between PNC, AGB and the SIs was more significant than 0.6 and 0.5, respectively (Table 4).

### 3.2. Estimation of Potato PNC and AGB Using Different Machine Learning Models

Figure 4 shows the PNC and AGB prediction results of different models without implementing data dimensionality reduction. The ELM could effectively improve the estimating accuracy of PNC and AGB (Figure 4f,l). However, the performance of the models was probably affected by the multicollinearity within input variables.

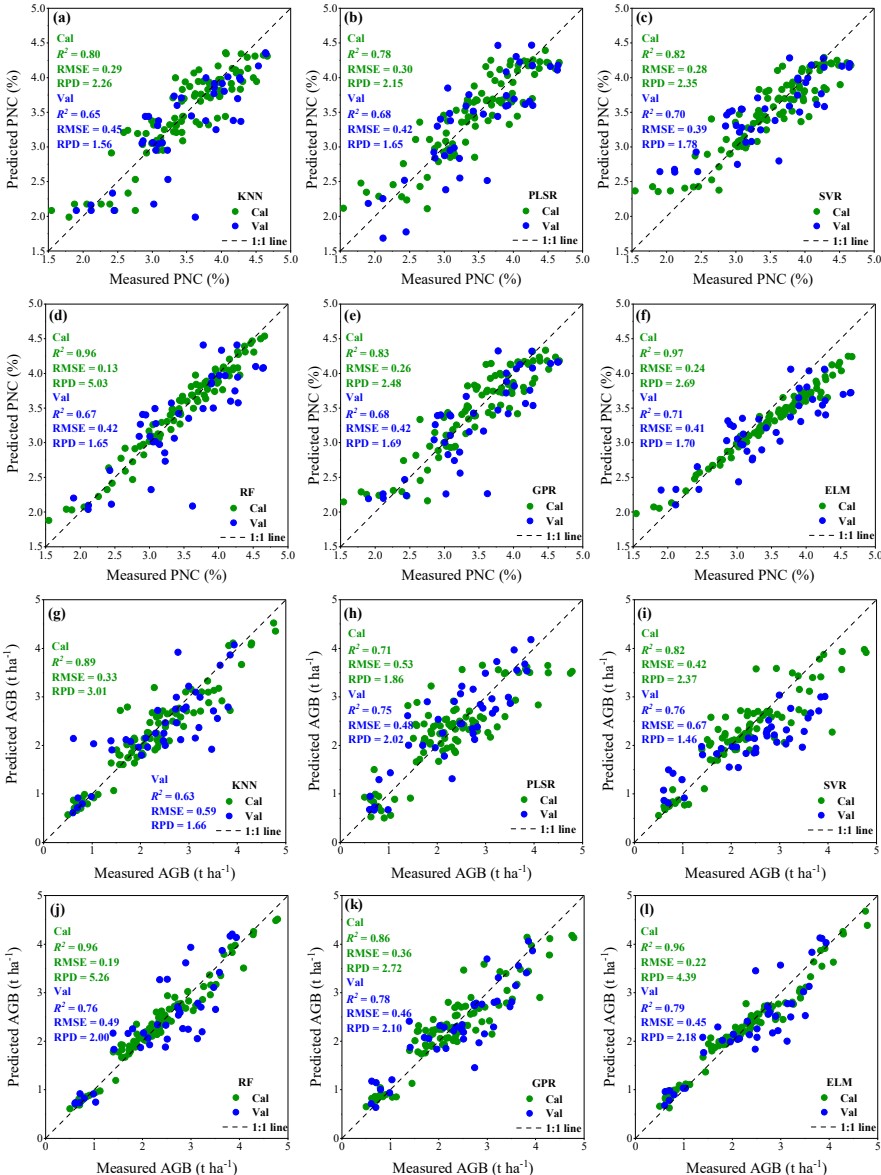

**Figure 4.** The performance of the estimation of six models for PNC (**a**–**f**) and AGB (**g**–**l**) without dimensionality reduction. Six models are as follows: K-nearest neighbor (**a**,**g**), partial least squares regression (**b**,**h**), support vector regression (**c**,**i**), random forest (**d**,**j**), Gaussian process regression (**e**,**k**), ensemble learning model (**f**,**l**).

### 3.3. Optimization of Input Variables of Machine Learning Models

#### 3.3.1. SPA and PCA-Based Data Dimensionality Reduction

Figure 5 shows the selected features based on the SPA with PNC and AGB. The input variables were chosen according to the RMSE. Figure 6 shows the accuracy of the PNC and AGB estimation models for all models based on the desired features. When SPA-based SIs were used as input variables, most prediction models were not significantly different from the original input variables (14 SIs as input variables), indicating that the SPA method could effectively reduce the number of SIs. The ELM had the best performance for PNC (Figure 6f) and AGB (Figure 6l) prediction compared to the other models.

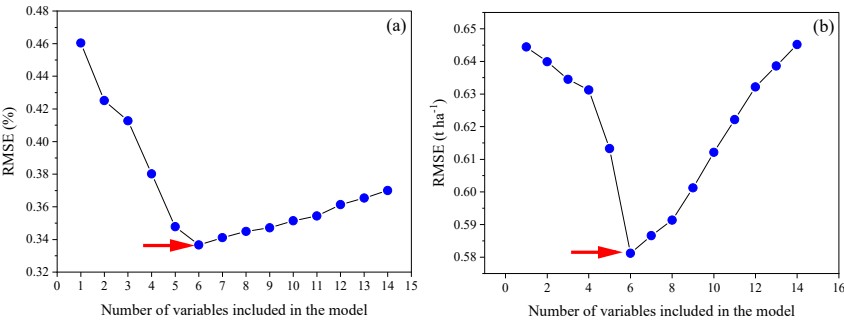

**Figure 5.** Input variables selection based on SPA algorithm. The red arrows represent the number of variables with the smallest RMSE (%). PNC: NDVI, RVI, MCARI, PSRI, NPDI, MCARI/OSAVI (**a**); AGB: OSAVI, MTCI, MCARI, PSRI, TCARI, MCARI/OSAVI (**b**).

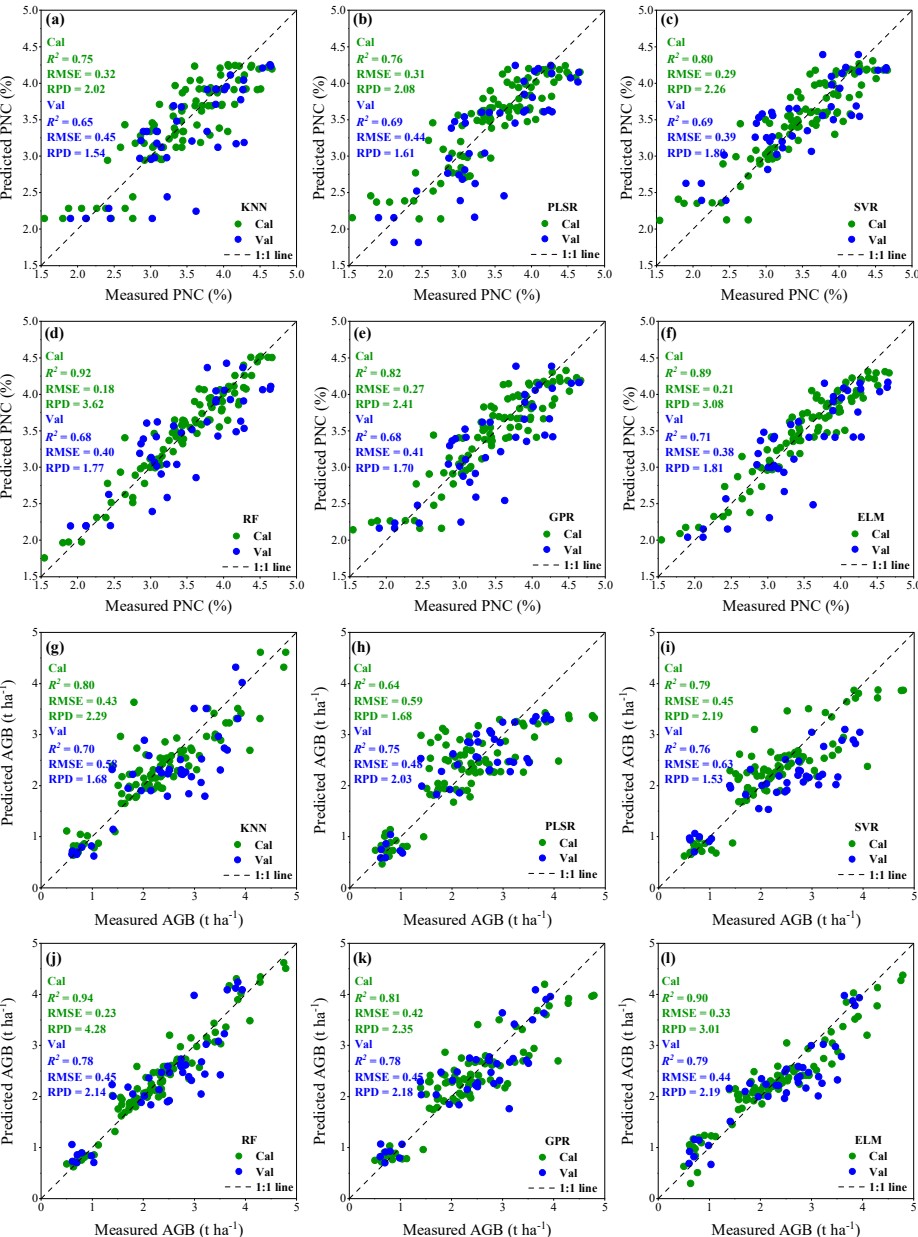

**Figure 6.** The performance of six SPA-based models: PNC (**a**–**f**); AGB (**g**–**l**). K-nearest neighbor (**a**,**g**), partial least squares regression (**b**,**h**), support vector regression (**c**,**i**), random forest (**d**,**j**), Gaussian process regression (**e**,**k**), ensemble learning model (**f**,**l**).

Figure 7 shows the processing of the PCA. The cumulative contribution of the first three PCs was more significant than 95% and can be used for model training. Similar to the SPA method, the performance of the models with the PCA-based SIs as input variables was not significantly different compared to the original input variables (Figure 8). The ELM performed best for potato PNC (Figure 8f) and AGB (Figure 8l) estimation.

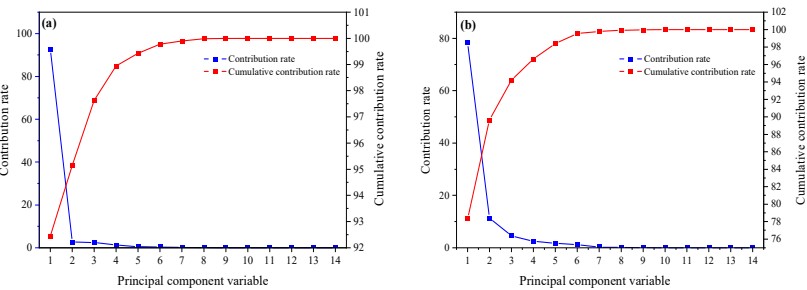

**Figure 7.** Input variables selection based on PCA algorithm. PNC (**a**), AGB (**b**).

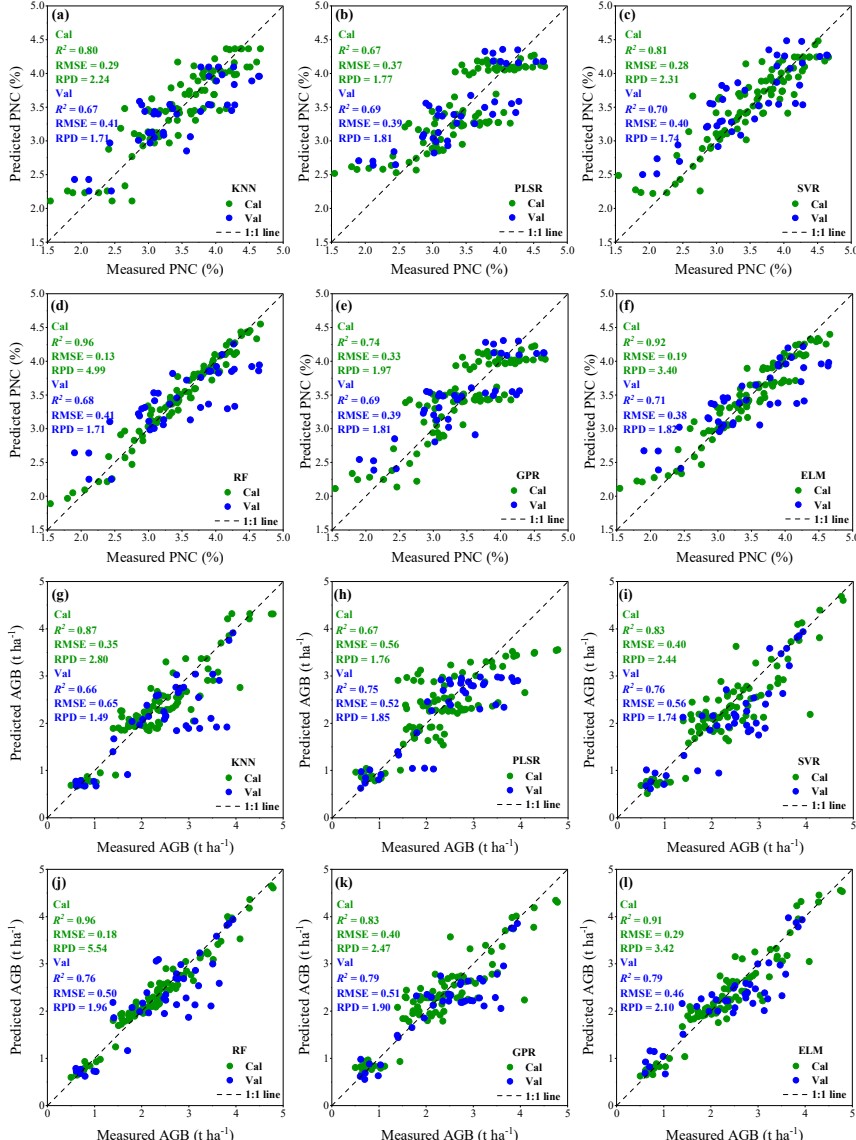

**Figure 8.** The performance of six PCA-based models: PNC (**a**–**f**), AGB (**g**–**l**). K-nearest neighbor (**a**,**g**), partial least squares regression (**b**,**h**), support vector regression (**c**,**i**), random forest (**d**,**j**), Gaussian process regression (**e**,**k**), ensemble learning model (**f**,**l**).

### 3.3.2. SPA-PCA-Based Data Dimensionality Reduction

Figure 9 shows the accuracy of the PNC and AGB prediction models established with six models and four input datasets. The results showed that SPA-PCA slightly improved with the PNC and AGB estimation models for most of the basic models for SIs. Compared with other models, the ELM that combined SPA-PCA-based SIs greatly improved the prediction accuracy of PNC (Figure 9a–c) and AGB (Figure 9d–f).

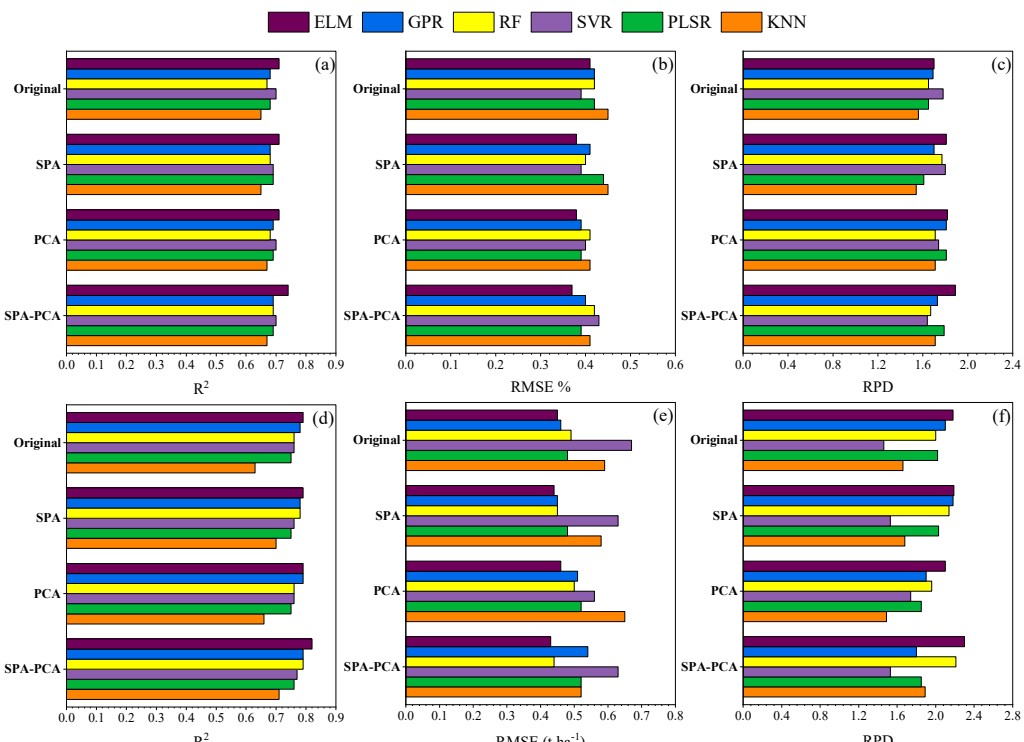

**Figure 9.** $R^2$, RMSE, and RPD of six models for potato PNC (**a**–**c**) and AGB (**d**–**f**) estimation. Original: 14 SIs as input variables; SPA: SPA-based input variables; PCA: PCA-based input variables; SPA-PCA: input variables based on SPA-PCA; K-nearest neighbor (KNN); partial least squares regression (PLSR); support vector regression (SVR); random forest (RF); Gaussian process regression (GPR); ensemble learning model (ELM).

### 3.4. Mapping PNC and AGB Using Sentinel-2 Imagery

The mapping results based on the ensemble learning model from the SPA-PCA are shown in Figure 10. The potato PNC was highest during tuber formation. With potato growth developed, the PNC of the potato was affected by the dilution effect [54] with a gradual decrease in PNC. In contrast to PNC, the AGB values were minimal during tuber formation. With potato growth developed, AGB gradually increased. In addition, Figure 11 demonstrates the correlation of PNC and AGB with yield at different potato growth stages. The results showed that biomass at the tuber-formation stage (T1) was highly correlated with potato yield (Figure 11b). Therefore, it is important to accurately monitor PNC and AGB during the critical growth stages of potato.

All measured PNC and AGB samples acquired synchronously with Sentinel-2 imagery were used for validation. Figure 12 shows the relationships between measured and predicted PNC and AGB. The accuracy of the prediction map is consistent with that of the spectral data prediction (PNC: $R^2$ = 0.72; RMSE = 0.46%; RPD = 1.46—AGB: $R^2$ = 0.85; RMSE = 0.37%; RPD = 2.64). This result also illustrates the accuracy of the prediction map.

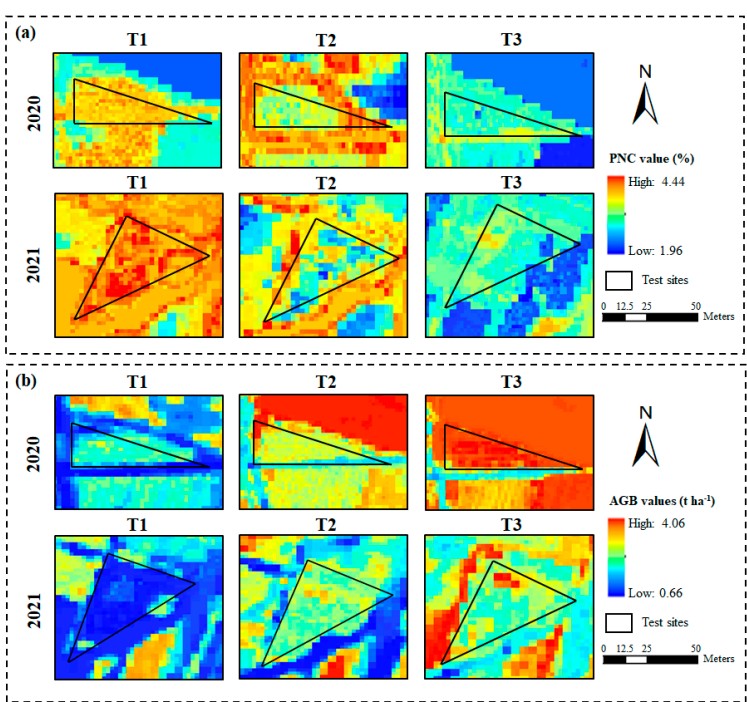

**Figure 10.** Spatial distribution of the estimation of PNC (**a**) and AGB (**b**) based on ELM method at different growth stages (T1: tuber-formation stage; T2: tuber-bulking stage; and T3: starch-accumulation stage).

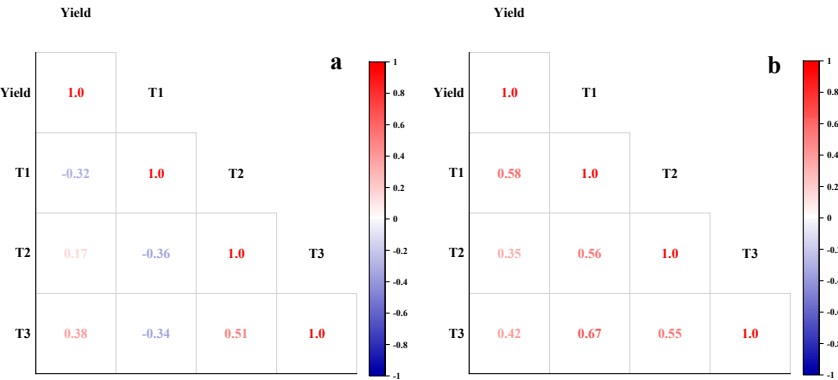

**Figure 11.** Correlation analysis of potato yield (t ha$^{-1}$) with PNC (**a**) and AGB (**b**) during different growth stages (T1: tuber-formation stage; T2: tuber-bulking stage; and T3: starch-accumulation stage).

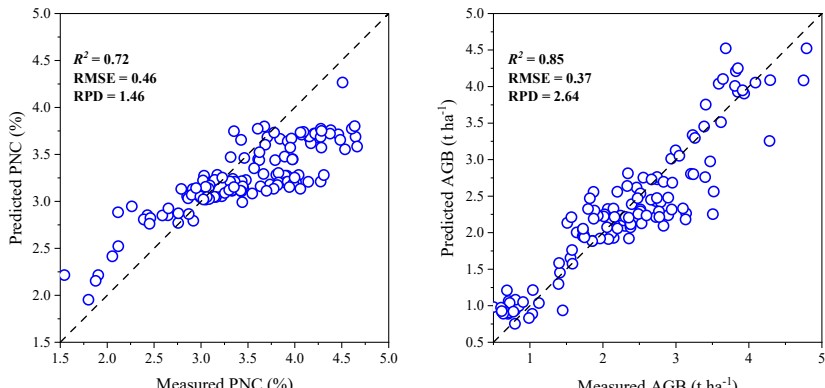

**Figure 12.** Relationship between measured and estimated PNC and AGB. PNC and AGB values estimated using the ensemble learning model with the optimal input variables.

## 4. Discussion

### 4.1. Comparison of PNC and AGB Sensitive Bands

Identifying and extracting critical bands from Sentinel-2 imagery is crucial for enhancing the accuracy of estimating crop PNC and AGB [55,56]. Among the tested SIs, the RE region and the SWIR were the most influential for estimating PNC and AGB (Table 4). Similar to the results of most studies, the RE is an effective band for monitoring crop parameters [57,58]. The RE is a region with sharp changes in reflectance, and is considered a sensitive region for monitoring the chlorophyll content of plants. The RE region changes depending on the strength of chlorophyll absorption [54]. However, the results in this study also indicate that SWIR can effectively improve the prediction accuracy of PNC and AGB (Table 4). Similarly, Perich [59] suggested that the SWIR was the best region for plant N status. This may be because protein and amino acid molecules in plants contain N. Proteins typically absorb short-wave infrared light. When using the SWIR to monitor plant growth, light in this band is absorbed by proteins in the plant, resulting in absorption peaks in the spectrum. Therefore, combining the SWIR band and RE can effectively monitor the potato PNC. The SWIR is also more sensitive to AGB. One possible explanation is that the highest aboveground biomass was achieved in the potato post-flowering (T2 + T3 stages) when the dilution effect diminishes [60]. The selected sensitive bands were weakly related to potato AGB during the post-flowering stages (Table 5). Thus, PNC has a more significant influence on choosing AGB sensitive bands. The results reported by Li [54] for winter wheat further confirm the more robust relationship between SIs and PNC when the biomass reaches constant values. Therefore, this study demonstrated the importance of the SWIR for potato PNC and AGB. At the same time, future research should also optimize the sensitive bands of PNC and AGB during different growth stages.

**Table 5.** Correlation coefficient between SIs and PNC and between SIs and AGB across different growth stages *.

| Spectral Indices | PNC | | AGB | |
|:---:|:---:|:---:|:---:|:---:|
| | T1 | T2 + T3 | T1 | T2 + T3 |
| NDVI | −0.31 | 0.56 | 0.26 | 0.02 |
| RVI | −0.31 | 0.52 | 0.22 | 0.03 |
| DVI | 0.49 | 0.65 | −0.77 | −0.39 |
| CI$_{red\ edge}$ | −0.31 | 0.52 | 0.22 | 0.03 |
| OSAVI | −0.10 | 0.58 | 0.42 | −0.41 |
| MTCI | 0.27 | −0.71 | 0.69 | −0.33 |
| MCARI | 0.49 | 0.65 | 0.90 | 0.39 |
| PSRI | 0.22 | −0.59 | 0.46 | −0.42 |
| mSR705 | −0.27 | 0.71 | −0.69 | 0.33 |
| mND705 | 0.29 | −0.65 | −0.59 | 0.02 |
| TCARI | 0.50 | 0.67 | −0.88 | −0.43 |
| NPDI | −0.29 | 0.66 | −0.31 | 0.03 |
| MCARI/OSAVI | 0.35 | 0.72 | −0.83 | −0.43 |
| TCARI/OSAVI | 0.35 | 0.72 | −0.83 | −0.43 |

* T1: tuber-formation stage; T2: tuber-bulking stage; and T3: starch-accumulation stage.

### 4.2. Comparison of Different Types of SIs

Although a variety of SIs have been successfully used to monitor crop PNC and AGB [61–63], limited research has evaluated the effectiveness of different types of SIs, such as chlorophyll-based, nitrogen-based, and biomass-based SIs, for the estimation of PNC and AGB of potato. In this study, four types of SIs were selected, including biomass, vegetation, chlorophyll/LAI, and nitrogen. The results showed chlorophyll-based SIs strongly correlate with potato PNC and AGB (Table 4). This is because chlorophyll, the green pigment in plants, contains N. Plants synthesize chlorophyll by absorbing N from the soil. Therefore, the SIs based on chlorophyll content are closely correlated with PNC [21]. In addition, chlorophyll content often represents plant growth conditions, and plants with

high chlorophyll content usually have high biomass. Therefore, SIs based on chlorophyll content and AGB often correlate strongly. Another reason may be the different levels of N fertilizer treatments set up in this study. The different nitrogen supply greatly affected the plant chlorophyll content and, thus, this is reflected in the PNC and AGB of the potato crops. However, unlike the correlation analysis, the SIs selected based on the SPA method included other SIs with weaker correlations (e.g., NDVI and RVI) (Figure 5). The SPA method includes an iterative selection of input variables and picks different numbers of SIs. Weaker SIs may highlight stronger ones and improve the predictive performance of the model.

### 4.3. Differences in Model Performance

Various research studies have employed individual machine learning models and SIs to assess crop PNC and AGB [20,62,64]. However, using a single machine learning model to estimate PNC and AGB still has limitations [22]. This is due to the inconsistent effect of different single models on limited datasets. This study constructed an ELM that combined different models (KNN, PLSR, SVR, RF, and GPR) for PNC and AGB estimation. The ELM produced a better potato PNC and AGB estimate performance than traditional machine learning models for multiple growth stages (Figure 9). In addition, predicted and measured values based on the ELM were closer to the 1:1 line and had higher LCCC values than the five base models (Figure 13), significantly reducing the overfitting phenomenon. Like Yang [22], the stacking ensemble algorithm yielded the highest estimation accuracy and lowest overfitting for crop parameters. This may be because each base model produces high estimation accuracy (Figure 9), and combining the advantages of various models could improve the estimation accuracy of PNC and AGB under different growth conditions. Moreover, this study compared three ensemble learning strategies (bagging, boosting, and stacking). The results demonstrated the advantages of the stacking model for predicting potato PNC and AGB (Figure 14). Therefore, future studies should determine which base models maximize the effectiveness of ensemble learning models and explore the potential of varying ensemble learning strategies, such as bagging or boosting technology, in different crop parameter estimations.

In this study, Sentinel-2 images were used to combine ensemble learning models for the accurate estimation of potato PNC and AGB. This provides effective help with the use of remote sensing to guide N fertilizer application. Currently, many studies have used crop PNC or AGB prediction based on spectral techniques to guide N fertilizer application. For example, Peng [32] predicted potato PNC using spectral indices and random forest regression models and calculated nitrogen nutrient indices (NNIs) in combination with potato dilution curves. The predicted NNIs provided help with potato nitrogen fertilizer guidance. Therefore, in future studies, we will construct local potato dilution curves to calculate NNIs in combination with potato PNC and AGB prediction models constructed by Sentinel-2 and ensemble learning to achieve large-scale, rapid, and non-destructive potato N fertilizer management.

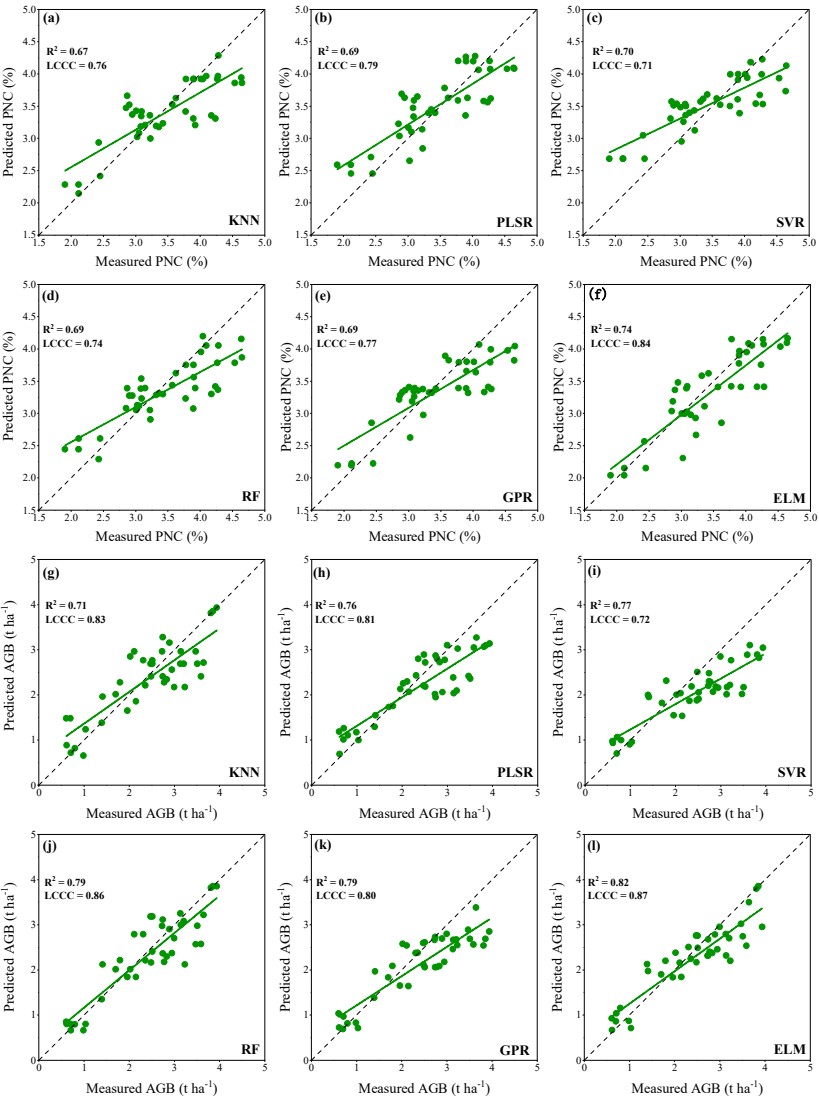

**Figure 13.** Evaluation of the performance of six models based on SPA-PCA. Six models: PNC (**a**–**f**), AGB (**g**–**l**). K-nearest neighbor (**a**,**g**), partial least squares regression (**b**,**h**), support vector regression (**c**,**i**), random forest (**d**,**j**), Gaussian process regression (**e**,**k**), ensemble learning model (**f**,**l**).

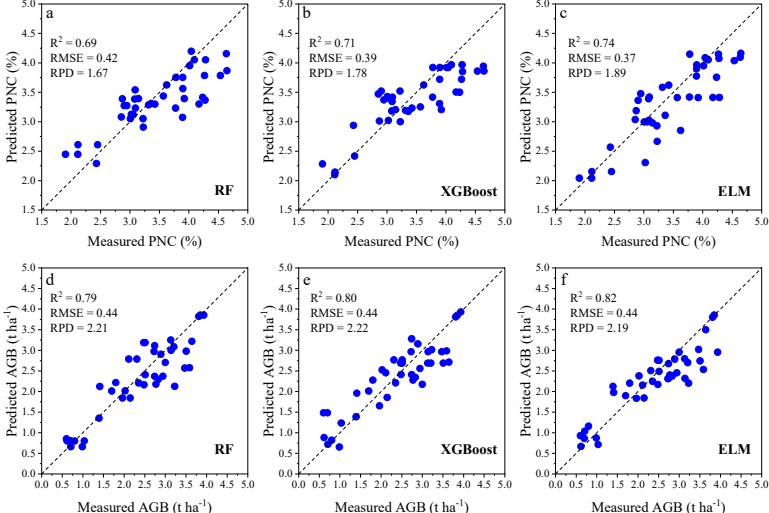

**Figure 14.** Evaluating the performance of RF (**a**,**d**), XGBoost (**b**,**e**), and ELM (**c**,**f**) based on SPA-PCA.

## 5. Conclusions

In this study, we compared the effectiveness of different types of SIs and different band combinations at predicting the PNC and AGB of potato. The SPA, PCA, and SPA-PCA were employed to select the optimal features to reduce multicollinearity among 14 SIs. The selected SIs were combined with six models (KNN, PLSR, SVR, RF, GPR, and ELM) for potato PNC and AGB estimation. The SIs based on chlorophyll content were strongly related to potato PNC and AGB. The sensitive bands were mainly concentrated in the red-edge (705 nm) and short-wave infrared (1610 and 2190 nm) regions. ELMs combined with SPA-PCA can effectively improve the prediction of PNC and AGB compared to other base models. The prediction performance was verified using ground-measured PNC and AGB data and the corresponding Sentinel-2 imagery. The resultant PNC and AGB maps confirmed the feasibility of predicting PNC and AGB with high-spatial-resolution Sentinel-2 imagery. This work highlights the potential value of Sentienl-2 data for the practical application of monitoring nitrogen use efficiency in agroecosystems.

**Author Contributions:** Experiments were designed by F.L.; H.Y. (Hang Yin), H.Y. (Haibo Yang) and Y.D. undertook the plant nitrogen concentration and aboveground biomass extractions in the field; H.Y. (Hang Yin) compiled the data and performed the machine learning analysis; H.Y. (Hang Yin) wrote the initial draft of the manuscript and F.L., Y.H. and K.Y. edited the manuscript. All authors have read and agreed to the published version of the manuscript.

**Funding:** This work was funded by the National Natural Science Foundation of China (32160757), Hohhot Applied Technology Research and Development Project (2023-Agriculture-5).

**Data Availability Statement:** The datasets used in this study are available on request from the corresponding author.

**Conflicts of Interest:** The authors declare no conflict of interest.

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
