# Peer review of "Mapping Plant Nitrogen Concentration and Aboveground Biomass of Potato Crops from Sentinel-2 Data Using Ensemble Learning Models"

_remotesensing, doi:10.3390/rs16020349_

Round 1
Reviewer 1 Report
Comments and Suggestions for Authors
This MS is Well-written, but some figure need to be improve. Also, please compare the accuracy if you don’t do PCA SPA. Add the introduction of ensemble method.
Major comments:
1. What is the specific experiment content? What is N1-N6 represent? You could give the specific content of references.
2. For Figure 4, maybe R2is better to understand, otherwise, if Pearson correlation is close to 0,99,so, single SI indices is good enough.
3. What is the specific model in figures a-f? The Figure f have highest accuracy. Maybe is the one after you use ensembled method.
4. Why do you use PCA,SPA method to reduce dimensions? Could you compare the accuracy if you didn’t do that. To my knowledge, the ensemble method, like random forest method, could avoid the” Curse of dimension”.
5. You also should intorduce which ensemble ML method you use. You can try and compare with XGboost,
Comments:
1. In figure1 the site images is too small;
2. The reason why you choose these 14 Spectral Indices? Please further explain in the text;
3. In section 2.5, what is usage in this research?
4. “The absolute correlation coefficient between the two parameters and SIs was more significant than 0.80 and 0.70.” What is the meaning of this sentence? Could you further explain?
Reviewer 2 Report
Comments and Suggestions for Authors
The introduction is clear and well-written. In general I found the methods to be described well, and the results and discussion were clear, though I had some small clarifying questions. I think the Discussion could be expanded slightly to address how PNC/AGB tracking could be used to guide N fertilization management, as is mentioned in the Introduction.
Line 70: Change “at scales” to “at large scales”.
Line 85: Change “while 90%-95% occurs from April to October” to “90-95% of which occurs from April to October”.
Lines 84-86: The June to September time frame would be more clear if you specifically state that is the potato growing season here.
Line 99: Change “tuber formulation” to “tuber formation”
Lines 100-101: The verb tense is not correct in this sentence. It should say something like “The sample was chopped and mixed, and a 400-600 g sample was taken.”
Lines 114-115: Were the bands resampled by nearest neighbor, bilinear interpolation, or another method? More explanation of this or the Sen2cor 2.5.5 software would be useful.
Lines 145-146: The model names should be fully spelled out here along with their abbreviation. For example, “K-nearest neighbor (KNN)”.
Lines 141-148: Can you provide any more detail about the ELM construction? Specifically, how are model predictions combined in this particular ELM? Are there any python packages that were used in this model? Are there citations of this ELM model you can provide?
Line 177: What do Ra, Rb, and Rc mean in Table 4? I’m confused by this table and why these numbers differ between PNC and AGB. Please describe in text and say what Ra, Rb, and Rc mean in the table caption.
Lines 226-232: Which model was used to predict PNC and AGB in the maps?
Line 241: Again, noting which model the predicted PNC and AGB were calculated from would be helpful here.
Reviewer 3 Report
Comments and Suggestions for Authors
Mapping Plant Nitrogen Concentration and Aboveground Biomass in Potato from Sentinel-2 Data Using Ensemble Learning Models (Remote Sensing ID: 2743508)
In this research, Sentinel-2 imagery and ensemble machine-learning techniques were employed to predict the plant nitrogen concentration (PNC) and aboveground biomass (AGB) of potato crops. The study identified sensitive spectral regions crucial for prediction analysis.
A few clarifications are warranted for a comprehensive understanding of this study:
-
1. Please explain the significance of PNC and AGB about the final yield of potato tubers.
-
2. Figure 11 illustrates the in-field variability of PNC and AGB. Does this variability impact the yield? Please include a section on the correlation between yield and PNC/AGB.
-
3. The methodology should provide a rationale for concentrating the analysis exclusively on potato tuber formation (T1), tuber bulking (T2), and starch accumulation (T3).
-
4. Table 3 should substitute the use of 'a,' 'b,' and 'c' with actual wavelength values (e.g., a- 430 nm)
-
5. Line 88 mentions the planting of forty-four plots, each with an area of 120 m2. Please clarify whether each experimental plot (N1-N5) correlates independently with image pixels. If so, considering a pixel size of 100 m2 against a plot size of 120 m2, potential misalignment issues may arise.
-
6. Additionally, clarification is needed on how PNC variability is managed, particularly if the average pixel value of the entire experimental plot is used.
7. Table 4 presents some confusion, particularly in the first row of NDVI, where spectral regions other than 660 (Red) and 860 (NIR) are mentioned. A clarification on this aspect is necessary.
Minor Comments:
- 1. The figure captions appear to be concise. Enhancing them with more details and ensuring they are self-explanatory would improve their overall clarity.
Round 2
Reviewer 3 Report
Comments and Suggestions for Authors
Accept in present form